# Development of Molecular-Based Species Identification and Optimization of Reaction Conditions for Molecular Diagnosis of Three Major Asian Planthoppers (Hemiptera: Delphacidae)

**DOI:** 10.3390/insects14020124

**Published:** 2023-01-25

**Authors:** Md-Mafizur Rahman, Hwayeun Nam, Nakjung Choi, Juil Kim

**Affiliations:** 1Agriculture and Life Sciences Research Institute, Kangwon National University, Chuncheon 24341, Republic of Korea; 2Department of Biotechnology and Genetic Engineering, Faculty of Biological Science, Islamic University, Kushtia 7003, Bangladesh; 3Incheon International Airport Regional Office, Animal and Plant Quarantine Agency, Incheon 22382, Republic of Korea; 4National Institute of Crop Science, Rural Development Administration, Wanju 55365, Republic of Korea; 5Program of Applied Biology, Division of Bio-Resource Sciences, College of Agriculture and Life Science, Kangwon National University, Chuncheon 24341, Republic of Korea

**Keywords:** loop-mediated isothermal amplification, multiplex PCR, planthopper, leafhopper, molecular diagnosis

## Abstract

**Simple Summary:**

The three planthoppers, brown planthoppers (*Nilaparvata lugens*), white-backed planthoppers (*Sogatella furcifera*), and small brown planthoppers (*Laodelphax striatellus*), are major economically important pests of rice. However, conventional morphology-based methods are inefficient in differentiating these pests at a species level. In this study, we report the successful design of species-specific primers and their use in general and multiplex PCR and loop-mediated isothermal amplification (LAMP) assays, the widely used tools for molecular species diagnosis. Each primer comprised a species-specific sequence of at least 2-mer or more in both forward and reverse primers. The primers displayed high diagnostic efficiency in conventional and multiplex PCR. Furthermore, the LAMP performed at 61 °C for 30 min using the inner, loop, and species-specific primers confirmed the reactions and successfully identified the species. Here, we used the DNA-releasing technique (reported elsewhere) to obtain the DNA from the tissue samples of the insects. The DNA yield was sufficient to perform the conventional and multiplex PCRs and LAMP assays. Additionally, we demonstrate the high sensitivity of LAMP to achieve positive amplification using from 100 pg to 10 pg genomic DNA. Overall, the techniques can effectively diagnose the planthoppers in a large number of field-collected or individual samples.

**Abstract:**

Asian planthoppers (Hemiptera: Delphacidae) that include brown planthoppers (BPH, *Nilaparvata lugens*, Stål), white-backed planthoppers (WBPH, *Sogatella furcifera*, Horváth), and small brown planthoppers (SBPH, *Laodelphax striatellus*, Fallén) are the primary sucking-type pests of rice. These three insects share morphological and sequence similarities. As insecticide resistance patterns and control strategies vary according to species, the accurate discrimination of these species is important. Here, we developed six species-specific primers based on partial mitochondrial genome sequences. The primers were successfully used in multiplex PCR, loop-mediated isothermal amplification (LAMP) assays, and conventional PCR. Here, we used genomic DNA obtained using the DNA-releasing technique (tissue samples were incubated at 95 °C for 5 min with 30 μL nuclease-free water, and the supernatant was used). We showed that multiplex PCR could analyze the density of each species following a mass collection in the field; the LAMP assay can diagnose the species within 40 min; conventional PCR can be widely applied to a large number of field samples, as well as individuals or mass collections. In conclusion, these results demonstrate the potential of the species-specific primers and DNA-releasing technique for accurate multiplex PCR and LAMP assays, which may assist the intensive field monitoring of integrated management of these species.

## 1. Introduction

Rice (*Oryza sativa* L.) is the most common cereal food in Asia and the Pacific, and more than 90% of global rice production is contributed by Asia [1,2,3]. The Asian planthoppers (Hemiptera: Delphacidae) that comprise *Nilaparvata lugens* (Stål, 1854) (brown planthopper, BPH), *Laodelphax striatellus* (Fallén, 1826) (small brown planthopper, SBPH), and *Sogatella furcifera* (Horváth, 1899) (white-backed planthopper, WBPH) are the most economically valuable sucking-type insect pests of rice [1,4,5]. They can cause significant losses in yield both by directly removing phloem sap and indirectly by spreading viral diseases, costing billions of USD annually [1,4]. Recently, the outbreaks of planthoppers have intensified across Asia [6], which is mainly attributed to the extensive use of broad-spectrum insecticides [7] and rarely to environmental factors, such as global warming [7,8]. Furthermore, morphological polymorphism, such as two different wing forms (dimorphic planthoppers), the phenotypic trade-offs between dispersal and reproduction capacity [9], a positive correlation between survival and fecundity [10], and patterns of diapause [11] are also important factors contributing to its increased infestation.

BPH was first introduced in the southern part of Korea and has recently been identified as the key pest of rice in Korea [12]. Since 1912, BPH has been recorded four times, with an estimated ten outbreaks. The severity of BPH has increased over the last decade, although it varies depending on the season [13]. Its infestation has also been recorded in other Asian countries, including Bangladesh [14], Japan [15], Thailand [16], and Vietnam [17]. The magnitude and timing of their movement influence the abundance of rice planthopper species in these temperate areas [18]. For the past five years (2011–2015), infestation by migratory SBPH from China has been reported during May and August [19]. Furthermore, the three rice planthoppers frequently co-occur in the same paddy field [20], and each has independently developed resistance to various insecticides [20,21,22,23]. In most cases, BPH is the biggest threat of the three planthoppers, while the other two are usually less destructive [16,21]. Most planthopper and leafhopper identification methods rely on morphological characteristics [24]. However, the high morphological similarities, distinct color variation, and the lesser differences in sizes at the nymph stage make it challenging to differentiate the three rice planthopper species using conventional morphological identification methods [25]. Therefore, identifying each rice planthopper species is necessary to efficiently manage these species at the beginning of their outbreaks in paddy fields. With the aim of the successful identification of the insects at the species level, several studies have employed molecular techniques that are not limited to the developmental or adult stages of the target planthopper species, such as restriction fragment length polymorphism-based conventional polymerase chain reaction (PCR) [26], multiplex PCR [27], and loop-mediated isothermal amplification (LAMP) assays [28,29]. Conventional PCR has been employed to identify the planthopper species; however, it can identify only one species at a time. In contrast, a multiplex PCR assay is useful for detecting more than two species [27]. Furthermore, as demonstrated in our previous study [30], the LAMP assay is useful for field diagnosis without genomic DNA (gDNA) isolation. LAMP is a highly specific technique, and its high sensitivity, speed, cost-effectiveness, reproducibility, and detection ability are the most important characteristics for such diagnostic assays. Some of the previous studies addressed the identification of three species [2], while in other studies, only one or two species were identified [28]. However, given the difficulties of sequence similarity [31,32] of the three species and the sensitivity of the PCR conditions [31], it is essential to develop appropriate primers and optimize the reaction conditions for the successful identification of the insects at the species level. Furthermore, to date, there have been no reports of using species-specific diagnostic primers in general PCR, multiplex PCR, and LAMP.

Mitochondrial DNA (mtDNA) offers a useful marker due to its low recombination rate, simple genetic structure, maternal inheritance, and relatively rapid rates of evolution [33,34]. The majority of studies have used multiple copies of mtDNA genes to generate species-specific primers. As a result, the goal of this study was to develop three species-specific diagnostic primer sets based on mt-genome sequences for BPH, SBPH, and WBPH. These primer sets were used for conventional PCR, multiplex PCR, and LAMP for various purposes, including single-, multiple-, and field-level planthopper species identification, respectively.

## 2. Materials and Methods

### 2.1. Sample Collection

Planthopper samples (brown planthopper (BPH; *Nilaparvata lugens*, Stål), small brown planthopper (SBPH, *Laodelphax striatellus*, Fallén), and white-backed planthopper (WBPH; *Sogatella furcifera*, Horváth)) were collected using the sweeping method from 12 locations across South Korea in between the months of June and August of the years 2021 and 2022 (Appendix A). The morphology of the three planthoppers and leafhoppers was observed using a Leica M205C microscope (Leica, Wetzler, Germany) and photographed with a flexacam c1 equipped with Leica Application Suite X (Leica). The morphological features of BPH differ from those of SBPH and WBBPH. For instance, jumping T3 (third thoracic segment) legs and long/short wing patterns are two distinct signs of BPHs [35].

### 2.2. DNA Sample Preparation and Primer Design

The gDNA of the three target planthoppers was directly extracted with DNAzol (Molecular Research Center, Cincinnati, OH, USA) based on the manufacturer’s protocol and quantified using Nanodrop (NanoDrop Technologies, Wilmington, DE, USA). Universal primers (LCO1490 and HCO2198) were used for species identification by sequencing following a previously described method [36]. gDNA isolated from each morphologically identified individual planthopper species was used as a template in a 20 μL PCR reaction containing 1 U TOYOBO KOD—FX neo (Toyobo Life Science, Osaka, Japan), 2× PCR buffer for KOD FX Neo (with 15 mM MgCl_2_), 0.2 mM each dNTP, 0.5 μM each primer, and 30 ng gDNA. After gel electrophoresis, the PCR products were directly sequenced (Macrogen, Seoul, Korea) and compared to the deposited genes using NCBI, BLASTN2.2.31+ [37] analysis. Sequence-confirmed DNA samples were used as standard/reference species (BPH, SBPH, and WBPH) samples. The GenBank accession number of each of the three-standard species (BPH, SBPH, and WBPH) was assigned to the following numbers: OP895722.1–OP895724.1, respectively. The species level could not be accurately identified using universal markers (mtCOI) [36]. After morphological identification, gDNA from ten individual planthoppers was mixed and used for PCR amplification with three biological replicates for each standard species sample. Finally, multiplex- and LAMP–PCR were used to confirm the species-level identification of each planthopper species accurately. DNA was extracted from confirmed BPH, SBPH, and WBPH samples and used as standard DNA for further PCR amplification.

The mitochondrial genomes of six Nilaparvata species (LC461184.1, MK590088.1, JX880069.1, NC_021748.1, JN563998.1, and KC333655.1), three Laodelphax species (NC_013706.1, FJ360695.1, and MK292897.1), and seven Sogatella species (MK907866.1, KC512914.1, KC512915.1, NC_021417.1, MW009064.1, NC_056127, and NC_042180.1) were downloaded from NCBI GenBank (https://www.ncbi.nlm.nih.gov/nuccore/, accessed on 26 November 2022). The sequences used in this study for designing the primers were mainly reported for planthoppers identified in Asian countries, including Korea and China. To design species-specific primers, Clustal W version 2.0 (UCD Conway Institute, University College Dublin, Ireland) was used to align the sequences with the default parameters [38]. MEGA version 11.0 was used to trim the ragged ends and phylogentic analysis [39]. Primers were designed using Premier 5 (Premier Biosoft International, Palo Alto, CA, USA), targeting the three planthopper sequences with some modifications.

### 2.3. Development of Multiplex PCR and LAMP

The PCR primers were validated and designed to amplify DNA fragments no longer than 700 bp to ensure that all multiplex PCR systems are well suited for tested insects with samples collected from only legs/antenna/whole body. After testing individual primer pairs in singleplex reactions to ensure proper amplification, optimal annealing temperatures were determined using PCRs, and primer concentrations were adjusted to balance the successful amplification of all fragments using standardized DNA templates, as described by Yashiro et al. [2]

TOYOBO KOD-FX neo (Toyobo Life Science) was used for general PCR to confirm the species through sequencing and primer checking. Appropriate primers were used with the PCR amplification protocol that included 2 min denaturation at 94 °C, followed by 40 cycles of denaturation at 94 °C for 15 s, annealing at 60 °C for 15 s, and extension at 68 °C for 15 s. SYBR green was used to visualize the amplified DNA fragments after they were separated on 1.5% agarose gel electrophoresis (Life Technologies, Grand Island, NY, USA). All experiments were conducted with at least three biological replicates.

The multiplex primer mixture comprised BPH_lamp3_F33/B33, SBPH_lamp1_F311/B34, and WBPH_lamp1_F31/B312 in a ratio of 1:1:1. A multiplex reaction (20 μL volume) was conducted with 3 μL of the respective primer mixture (10 pmol/μL), 1 μL gDNA (10 ng/μL), 10 μL 2X buffer with 15 mM MgCl_2_, 1.6 μL dNTP (each dNTP), including 0.15 KOD-FX neo (1 unit/μL), and PCR grade water 5.25 μL. The multiplex PCR reaction was conducted for 2 min at 94 °C, followed by denaturation at 94 °C for 20 s, annealing at 53 °C for 20 s, extension at 68 °C for 20 s, and final extension at 68 °C for 2 min, and holding at 8 °C for an unlimited period. Amplification was carried out using an Applied Biosystems ProFlex PCR system (Thermo Fisher Scientific, Waltham, MA, USA).

The LAMP assay was carried out according to the manufacturer’s instructions using a WarmStart^®^ LAMP Kit (New England Biolabs, Ipswich, MA, USA). The LAMP conditions were selected based on our previous studies [30,40]. The LAMP assay was run for 60 min at 61 °C, 63 °C, and 65 °C with four primers (F3, B3, FIP, and BIP) to optimize the reaction temperature using an Applied Biosystems ProFlex PCR system (Thermo Fisher Scientific). The efficiency of loop primer(s) was tested/evaluated in the presence of additional loop primer(s) at 61 °C for 30 min. The detection limit of gDNA was also tested using six primers at 61 °C for 30 min.

### 2.4. Field Application

To validate species-specific PCR, we collected at least four samples from each (ten different localities) region of South Korea (Appendix A). gDNA was isolated, quantified, and used for amplification of the target sequences using the species-specific primer sets for BPH (B1–B5), SBPH (S1–S5), and WBPH (W1–W5). Forty samples were randomly tested for validation and confirmation using each primer pair. For the LAMP, the DNA was prepared using the DNA-releasing technique [40]. Briefly, tissue samples obtained from the whole body or some body parts, such as the antenna or leg, were incubated at 95 °C for 5 min with 30 μL nuclease-free water. After incubation, 2 μL supernatant (DNA source) was used for the LAMP and incubated in bench top type dry bath-heating block (MaXtable^TM^ H10, Daihan Scientific, Seoul, Korea) at 61 °C for 30 min using six primers.

## 3. Results

### 3.1. Primer Designing and Primer Selection

The multiple alignments of mitochondrial genome sequences to design species-specific primer sets (Figure 1 and Appendix A) revealed intraspecies SNPs.

The positions of the LAMP primers and primer-binding regions in partial mitochondrial genomes are presented in Figure 1. In total, 16 sequences were aligned to select species-specific regions for primer design. The primary diagnostic primers used were F3 and B3. The inner primer FIP comprised F1c (a complementary sequence of F1) and F2. Another inner primer, BIP, comprised B1 and B2c (complementary sequences to B2). The dumbbell structure was generated using four essential LAMP primers (F3, FIP, BIP, and B3), and the LAMP reaction was accelerated by two loop primers, namely loop forward (LF) and loop backward (LB). The primers used in this study are listed in Table 1 and primers positions are shown in Figure 1. SNPs and/or InDels in the priming region were confirmed using the strand DNA samples of each species collected in Korea via sequencing (over 20 individual samples of each species were verified).

A species-specific region that can diagnose three different planthopper species (*N. lugens*, *L. striatellus*, and *S. furcifera*) was identified. First, we used single-plex PCR using the samples of all three planthoppers separately at 60 °C, 58 °C, 55 °C, and 53 °C; multiplex PCR was performed at 53 °C, 55 °C, 58 °C, and finally, 53 °C was optimized (Figure 2).

For the selection criteria of LAMP, 10 ng gDNA of three planthopper species, for which the species was confirmed using the developed primer set (four primers, F3, B3, FIP, and BIP for LAMP), was added and incubated at 61 °C for 1 h. For multiplex PCR, since there was no interference between primers, a primer set capable of accurately diagnosing the species was selected. Based on primary screening, the most effective primer sets were selected for each insect species (Table 1). In the targeted three planthopper species diagnostic primer sets, the best results were obtained at 61 °C among three different temperatures (65 °C, 63 °C, and 61 °C) (Figure 3). The primers were then chosen to definitively identify the species.

In Appendix A, we amplified each of the designed primer sets, but some of them produced ambiguous, double bands with the target band, or no band with the tested planthopper samples. Based on the results, set 3 was selected for BPH (NADH dehydrogenase subunit 1), set 4 for SBPH (NADH dehydrogenase subunit 2), and set 1 for WBPH (NADH dehydrogenase subunit 2) (Appendix A and Figure 2). The selected primers were used for LAMP and multiplex PCR (Figure 2). Amplification of the samples from the leafhoppers was used as an outgroup, and non-template DNA was used as a negative control in all experiments performed to verify the species.

### 3.2. Comparison of Loop Primer Efficiency

We evaluated whether the use of the loop primers LB and LF could increase responses in addition to the four essential primers (F3, B3, FIP, and BIP) by adding each loop primer individually (LB or LF) or simultaneously (LB and LF). Except for BPH, where primer addition increased reaction efficiency, this resulted in non-specific reactions. When the two loop primers were added and reacted simultaneously, the reaction efficiency increased significantly. Figure 4 depicts representative electrophoresis images.

### 3.3. Determining the Diagnosis Limit Concentrations

For each planthopper species, the diagnostic concentrations were reduced by one-tenth from 100 ng to 100 fg in a 25 L standard reaction solution. The results indicate that BPH and SBPH were successfully detected at concentrations of up to 10 pg, and WBPH was detected at concentrations up to 100 pg (Figure 5). These standards were determined based on the observations of the reaction mix under visible and ultraviolet lights and by performing gel electrophoresis. The identification results for the three types of planthopper insects are consistent, indicating that a concentration equal to or higher than that detected by any of these methods (single- or multiplex PCR) could be used to successfully amplify the targeted samples.

### 3.4. Field Application

For the field application, 40 samples were chosen at random from nine different locations (Wando, Jindo, Haenam, Muan, Hampyeong, Yeonggwang, Goechang, Donghae, and Chuncheon). We were unable to detect any BPH samples, which could be attributed to a sample scarcity, most likely due to seasonal variation [42]. However, the target bands for WBPH were amplified in 27.5% (11/40) of the samples, and those for SBPH were positive in 47% of the samples (19/40) (Appendix A). For the field samples tested, we used multiplex PCR and LAMP for specificity and validation of our primers.

The field application method was designed without the need for a separate DNA extraction process that can be used in conjunction with the LAMP method (Figure 6). A segment of the antenna or leg tissue from BPH, SBPH, and WBPH adults was cut and reacted at 95 °C for 5 min to release DNA, as previously reported for invasive lepidopteran pests such as *Spodoptera frugiperda* [43]. The DNA concentration was determined by measuring the amount of released DNA using a nanodrop system; however, the concentration varied depending on the amount of tissue in the initial reaction. Overall, enough DNA was released for the reaction, yielding the same results as the positive control (DNAzol, a standard method for isolating DNA from insect tissue, [44]) for BPH, SBPH, and WBPH. The results of all three methods (LAMP-, multiplex-, and conventional-PCR) were consistent, and gel electrophoresis of the PCR products confirmed the positive reactions. These results suggest that visual observations under visible light are preferable for confirming LAMP reactions and can be used as standard methods in the field.

## 4. Discussion

Morphological differences alone may not be sufficient to distinguish insects, particularly rice planthoppers (*N. lugens*, *S. furcifera*, and *L. striatellus*), which cause “hopper-burn” in paddy fields [45]. Moreover, conventional morphology-based identification requires taxonomic knowledge; however, there are no taxonomic keys for the nymph stage of planthoppers [26]. To overcome the limitations of morphological identification methods, several studies have explored the potential of DNA-based molecular markers to distinguish between insect species with indistinct morphological characteristics, such as cryptic species or immature life stages [46,47,48]. In particular, mitochondrial genes have been used as a source of target-specific primers in recent studies due to their high copy number and inherent resistance to degradation [30,43]. In this study, we developed field-level diagnostic LAMP, multiplex PCR, and conventional PCR markers based on the allelic polymorphism of aligned mitochondrial DNA sequences. Some studies [27,31] have attempted to identify these using the internal transcribed spacer (ITS) region of planthoppers. However, these methods have limitations, such as the need for an additional DNA extraction step. Nevertheless, no previous research on planthopper species identification based on mitochondrial DNA has been conducted. Here, we developed and validated the species-specific LAMP-, multiplex-, and conventional-PCR markers, each for a different purpose.

No matter how much the diagnostic primers were designed based on a large number of planthoppers’ sequence data available in GenBank and how the diagnostic method was developed, undoubtedly, this study also has limitations. First, due to the insufficient number of samples used (less than 100 individual samples per each species), particularly, the international local diversity of the local sample is too narrow. Second, insufficient SNP analysis, which is the confirmation of the partial mt genome sequence of species that may occur in other regions other than the main species in Korea, etc. However, despite these limitations, it is estimated that the method developed in this study can be widely utilized for early diagnosis of the three species of planthoppers, which are the most problematic in paddy rice fields not only in Korea but also in various other countries.

In point-of-care settings, a molecular diagnosis is an important tool for developing efficient insect pest management strategies [30]. Therefore, appropriate primer design and validation using sufficient samples are required. Finding suitable loop primers is difficult due to the high AT content, and the strategy of introducing additional overhanging ‘G’s to the 5′ end to enhance the annealing temperature is difficult to apply to the inner primers [40,43]. We added overhanging ‘G’s to the 5′ end of the developed LAMP primers (Table 1) and validated it in identifying the three planthopper species at the field level. Six primers, including two inner primers (FIP and BIP), were used in the LAMP reaction. The LAMP reaction assay is more sensitive to reaction temperature than conventional PCR but less sensitive than nested and quantitative PCR [49]; therefore, the optimization of reaction conditions based on the primers and samples used is important. In this study, we added a loop primer previously known to improve the PCR amplification efficiency [30,40] to develop the protocol with the highest sensitivity; we verified the amplification efficiency at different temperatures (65 °C, 63 °C, and 61 °C) and identified 61 °C as the suitable temperature. Moreover, the reaction time was reduced by 30 min with the two loop primers (LF and LB) while maintaining the highest amplification efficiency. However, LF was less efficient than LB (Figure 4). By adding each or both primers simultaneously, the additional loop primers could improve the reaction efficiency. In particular, when the two loop primers (LF and LB) were added and reacted simultaneously, the reaction efficiency was significantly improved; a similar observation was followed in our previous research [30]. In our previous study, we established a LAMP assay with a simple DNA-releasing technique [43], representing a valuable diagnostic tool in the field where instruments and reagents may be limited. Reportedly, the potential limitations and drawbacks of DNA concentration can be overcome without the use of DNA extraction (mentioned earlier). In this study, we used this simple technique to overcome the requirement of a DNA extraction step and successfully isolated the DNA with sufficient yield to perform the assays used in this study. Using six primers (four basic primers and two loop primers, LF and LB), the diagnostic DNA concentration limit was 10 pg for BPH and SBPH, and 100 pg for WBPH, which was consistent with a previous study [50]. Overall, when LAMP is applied in the field without a separate standard DNA extraction process, a body part (leg or antenna) from the planthopper insect is incubated in a reaction tube 95 °C for 5 min with 30 μL nuclease-free water and then the supernatant (2 μL) is used for LAMP reaction with the primer; thus, the developed LAMP assay can save time, money, and effort when identifying planthopper samples.

Multiplex PCR has many applications, including insect diagnosis [2], pathogen detection [51], and forensic testing [52]. Its simplicity of use and low setup cost has opened up new opportunities for its development. In this study, multiplex primers (combinations of three conventional primer sets: BPH, SBPH, and WBPH primer pairs) were used for the diagnosis of a large number of insect samples. Although a similar approach was used in a previous study [2], in our study, non-specific bands were observed when the annealing temperature was combined with the PCR conditions, indicating reduced amplification efficiency. This could be due to the sensitivity of multiplex PCR to reaction temperature and DNA concentration [2]. Therefore, we performed several experiments to confirm the optimal temperature and minimum DNA concentration of multiplex PCR. Finally, a suitable temperature of 53 °C and a DNA concentration of 10 ng/μL were selected for multiplex PCR (Figure 1). This DNA concentration was achievable by using the simple DNA-releasing technique and using the tissue samples of any body part of the insects.

Conventional PCR primers are used for gene-based diagnostic purposes [53]. Moreover, conventional PCR with a species-specific primer set is useful for identifying planthopper samples for high throughput analysis in lab. Here, we tested a large number of samples collected from 12 different locations (Appendix A). This can help identify the species even using gDNA extracted from a large number of field-collected samples and/or individual samples. We demonstrated the amplification efficiency of the species-specific primer sets using DNA samples collected using the DNA-releasing method. Depending on the purpose, various molecular approaches were employed for planthopper identification.

The LAMP technique is a simple, low-cost, rapid, highly selective, and sensitive process; however, it is critical to consider the amplification of false positives while using LAMP in diagnostic kits [54]. Visual methods, specifically those based on turbidity or color change, may not represent target-specific reactions; distinguishing true positives from false positives or non-target amplification is difficult. [54]. Similarly, multiplex PCR has some limitations, such as self-inhibition within the primer mixture and low efficiency compared to conventional PCR primer efficiency, particularly when using low DNA-quality samples. We overcame this problem, as mentioned in ‘Primer Designing and Primer Selection’ of the Section 3, and an optimal primer set was selected with different planthoppers and closely related leafhopper samples, considered as outgroup. Therefore, it was possible to apply to conventional PCR, multiplex PCR, and LAMP using one species-specific primer set. The reason is that each method has advantages and disadvantages so that the user can make an optimal diagnosis according to their own purpose.

## 5. Conclusions

In this study, we developed three species-specific primer sets for traditional PCR, multiplex PCR, and LAMP, all of which are commonly used for molecular species identification. Primers, specifically LAMP, can be used in the field, and their sensitivity is high enough to detect even small amounts of DNA (obtained by the DNA-releasing method). Multiplex PCR using the developed species-specific primer is useful for diagnosing a large number of insect samples. Furthermore, we also demonstrated the efficiency of conventional PCR for the diagnosis of a large number of field-collected samples and/or individual samples. Overall, the DNA-releasing technique combined with the reported species-specific primers derived from mitochondrial DNA sequences of the three major planthoppers and the optimized reaction conditions could be useful for the rapid and cost-effective identification of the insects; therefore, the findings could be useful for the development of strategies for the effective integrated management of this insect.

## Figures and Tables

**Figure 1 insects-14-00124-f001:**
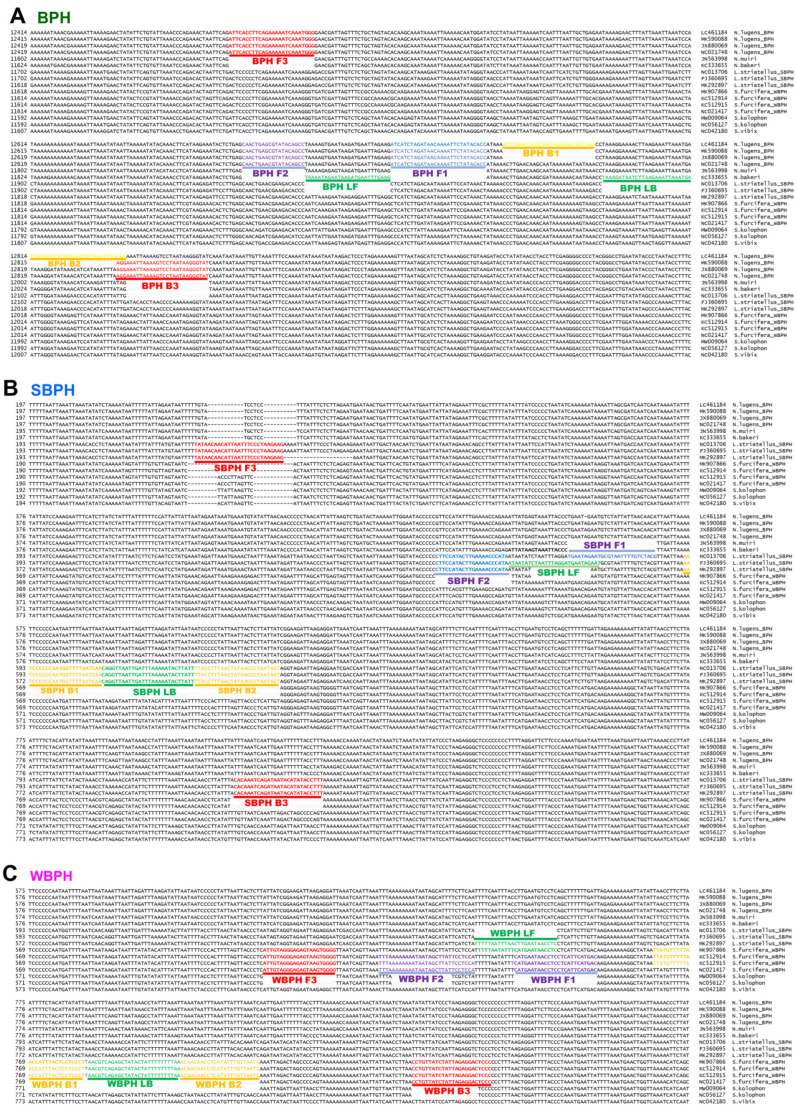
Location of primers and primer-binding regions on partial sequence of mitochondrial genomes of six *Nilaparvata*, three *Laodelphax*, and seven *Sogatella* species. (**A**). a diagram of the brown planthopper (*Nilavarpa lugens*), (**B**). a diagram of Small brown planthopper (*Laodelphax striatellus*), and (**C**). a diagram of White-backed planthopper (*Sogatella furcifera*). In total, 16 sequences were aligned to select a species-specific region for primer design. The primary diagnostic primers are F3 and B3. The inner primer, FIP, comprised F1c (complementary sequence of F1) and F2. Another inner primer, BIP, comprised B1 and B2c (complementary sequences of B2). The dumbbell structure was generated by four essential LAMP primers (F3, FIP, BIP, and B3), and the LAMP reaction was accelerated by two loop primers, namely LF and LB. BPH, brown planthopper; WBPH, white-backed planthopper; SBPH, small brown planthopper.

**Figure 2 insects-14-00124-f002:**
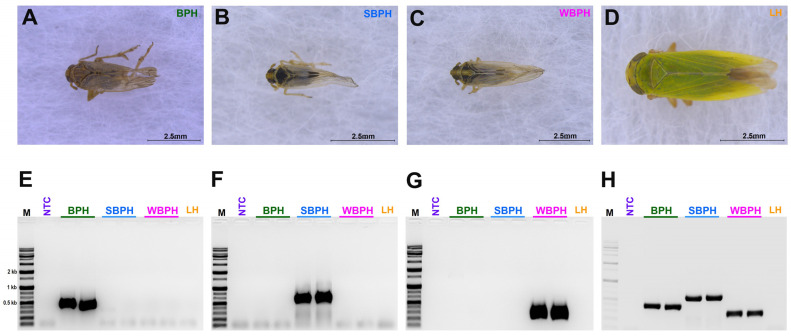
The morphological features of three tested planthopper samples (**A**) *Nilaparvata lugens,* BPH (**B**) *Laodelphax striatellus*, SBPH (**C**) *Sogatella furcifera,* WBPH (**D**) *Nephotettix nigropictus,* Leafhopper (LH) (outer group); (**E**–**H**) The single (**E**–**G**)- and multiplex (**H**) polymerase chain reactions (PCRs) conducted to identify BPH, SBPH, WBPH, and LH. BPH, brown planthopper; WBPH, white-backed planthopper; SBPH, small brown planthopper; LH, leafhopper.

**Figure 3 insects-14-00124-f003:**
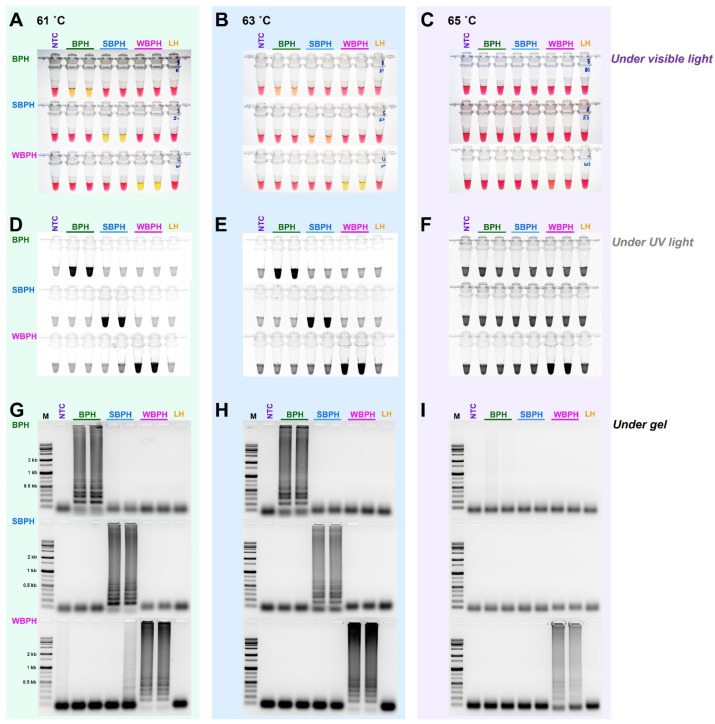
The sensitivity of the LAMP assay using the specific primers obtained by aligning 16 mitochondrial genome sequences of three planthoppers under three different temperatures (61 °C, 63 °C, and 65 °C). The reactions were visualized under (**A**–**C**) visible light, (**D**–**F**) ultraviolet light with SYBR green, and (**G**–**I**) by performing gel electrophoresis. When the product was formed, the existing pink color of the reaction mixture turned yellow in a strong response but remained pink in a negative response [40]. Magnesium ions react with pyrophosphate derived from dNTPs to form a precipitate during the LAMP reaction; the resulting depletion of magnesium ions in solution results in a color change, indicating a positive reaction [41]. LAMP assay was tested in an incubation temperature of 61 °C for 1 h. NTC: no template control, BPH: brown planthopper, SBPH: small brown planthopper, WBPH: white-backed planthopper.

**Figure 4 insects-14-00124-f004:**
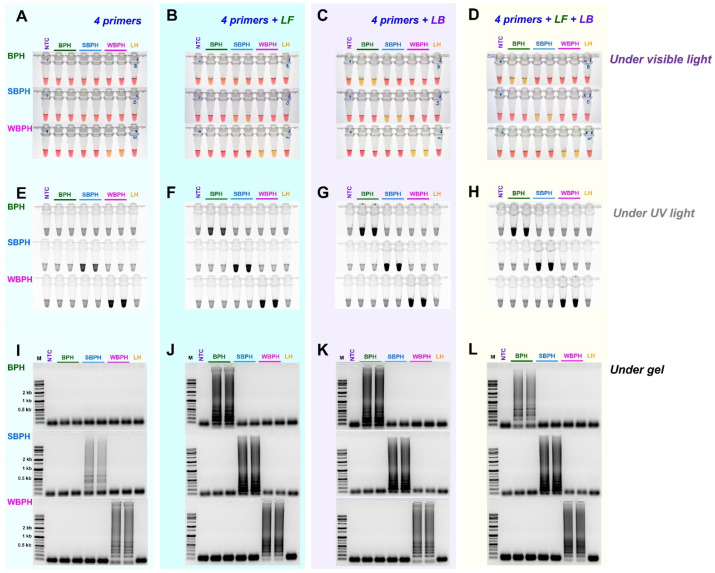
The efficiency of LAMP assay with four primers (F3, B3, FIP, and BIP) and additional loop primers, loop backward (LB), and loop forward (LF). LAMP assay was tested at an incubation temperature of 61 °C for 30 min, and the reactions were observed (**A**–**D**) under visible light, (**E**–**H**) under ultraviolet light after treating with SYBR green, and (**I**–**L**) by performing gel electrophoresis. NTC: no template control, BPH: brown planthopper, SBPH: small brown planthopper, WBPH: white-backed planthopper.

**Figure 5 insects-14-00124-f005:**
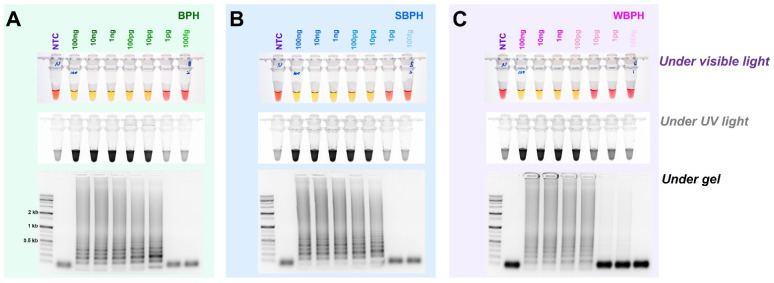
The detection limit of the concentrations of genomic DNA in the LAMP assay of BPH, SBPH, and WBPH samples. gDNA (100 ng–100 fg) from the planthoppers were used for the assay (**A**–**C**). The reactions were observed under visible light, ultraviolet light after treatment with SYBR green, and by performing gel electrophoresis. NTC: no template control, BPH: brown planthopper, SBPH: small brown planthopper, WBPH: white-backed planthopper.

**Figure 6 insects-14-00124-f006:**
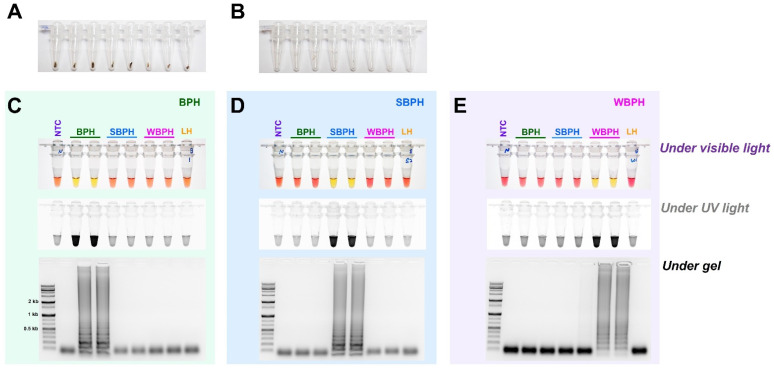
Field application of DNA-releasing technique (5 min incubation at 95 °C with 30 µL nuclease-free water, (**A**). whole insect (**B**). only leg/antenna) to detect the species-specific target bands amplified using gDNA of BPH (**C**), SBPH (**D**), and WBPH (**E**). LAMP assay results were carried out by incubating the reaction mix at 63 °C for 30 min with six primers (F3, B3, FIP, BIP, LF, and LB). The observations were recorded under visible light, ultraviolet light with SYBR green, and gel electrophoresis. NTC: no template control, BPH: brown planthopper, SBPH: small brown planthopper, WBPH: white-backed planthopper.

**Table 1 insects-14-00124-t001:** Primers used for LAMP, PCR, and multiplex PCR.

Insect Sample Name	Primers **	Sequence (5′–3′) *
Brown planthopper (BPH, *Nilaparvata lugens*, Stål)	BPH_lamp3_F33	ATTCACCTTCAGAAAAATCAAATG**GG**
	BPH_lamp3_B33	ATACCCTTATTAGGACTTTTAATTTC**CT**
	BPH_lamp3_FIP	GGTGTATAGAATTTTGTTATCTAGATGATCCAACTGAGCGTATACAGCC
	BPH_lamp3_BIP	ACTTGAACAAGCAATAAAAAATAATAAACCCCTTAAATTTATGATGTTTATATCCTT
	BPH_lamp3_LF	CTTCTAATTCATCTTATTCACTTTTA
	BPH_lamp3_LB	TAAAGGAAAACTTAAGAAATTAAATGATA
Small brown planthopper (SBPH, *Laodelphax striatellus*, Fallén).	SBPH_lamp1_F311	***GG***TATAACAACATTAATTTCCCT**AAGAAG**
	SBPH_lamp4_B341	***GG***AAAGGTATATGTATTATCTGATTT**GTG**
	SBPH_lamp14_FIP	CGTAGACAAAATTACGCATTCTATTCACTTCCATACTTGAAAACCCATA
	SBPH_lamp14_BIP	AATCCCCCCAATGGTTTTAATCAAACTATTAGGGTTATAGTTAAAGGTAA
	SBPH_lamp14_LF	ATTCTATTCATCCTAAATTAGATATTAT
	SBPH_lamp14_LB	CAGGTTAATTGATTTAAAAATACTTATT
White-backed planthopper (WBPH, *Sogatella furcifera*, Horváth),	WBPH_lamp1_F31	ATTGTAGGGAGAGTAAGTGG**GG**
	WBPH_lamp1_B312	GGGAGTCCTCTAATAGATAAC**AGG**
	WBPH_lamp1_FIP	GTCATGAATGAGGAGGTTATTCATGATTTAAAAAAAATAATAGCTTATTCCTCCAT
	WBPH_lamp1_BIP	TTATGTTTTTAACCATTTACTCCTCCCTGATTAACAAATATGAGGTTGTTGT
	WBPH_lamp1_LF	GAGGAGGTTATTCATGAAATATTAAAA
	WBPH_lamp1_LB	TAACGTCAGAGCTATACTATTTTTTTTAA
Planthopper and leafhopper identification	LCO1490	GGTCAACAAATCATAAAGATATTGG
	HCO2198	TAAACTTCAGGCTGACCAAAAAATCA

‘*’ The bold underlined bases are species-specific (allele-specific target bases of planthopper mitochondrial genome sequence). ‘**’ bold italicized bases are additional bases for annealing temperature adjustment.

## Data Availability

All of the available data are produced or analysis with this manuscript.

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
