# Peer review of "Development of Molecular-Based Species Identification and Optimization of Reaction Conditions for Molecular Diagnosis of Three Major Asian Planthoppers (Hemiptera: Delphacidae)"

_insects, 2023, doi:10.3390/insects14020124_

Round 1
Reviewer 1 Report
It is difficult to distinguish BPH, WBPH and SBPH in nymphs based the conventional morphological methods. In this study, the authors developed the successful designing of species-specific primers to use in general and multiplex PCR and loop-mediated isothermal amplification (LAMP) assay, which can effectively diagnose the planthoppers in a large number of field-collected or individual samples.
I think the topic is an orignial work, mainly the successful development of species-sppecific primoers to for the identificiation of three insects. And this work simplifies the diagnose of three insects species in the flied.
I have no additioanl comments.
Author Response
Reviewer 1 comments and responses:
Comment 1: English language and style are fine/minor spell check required
Response 1: We carefully checked and updated the whole manuscript. The native speaker also checked the manuscript.
General Comments 2:
It is difficult to distinguish BPH, WBPH and SBPH in nymphs based the conventional morphological methods. In this study, the authors developed the successful designing of species-specific primers to use in general and multiplex PCR and loop-mediated isothermal amplification (LAMP) assay, which can effectively diagnose the planthoppers in a large number of field-collected or individual samples.
I think the topic is an orignial work, mainly the successful development of species-sppecific primers to for the identification of three insects. And this work simplifies the diagnose of three insect’s species in the flied.
I have no additional comments.
Response 2: Thanks for your positive comments regarding the manuscript.
Reviewer 2 Report
This work developed three species-specific primer sets for traditional PCR, multiplex PCR, and LAMP to distinguish insects, particularly rice planthoppers (N. lugens, S. furcifera, and L. striatellus). I think this work is interesting and useful.
Author Response
Reviewer 2 comments and responses:
Comment 1: English language and style are fine/minor spell check required
Response 1: We carefully checked and updated the whole manuscript. The native speaker also checked the manuscript.
Comment 2: This work developed three species-specific primer sets for traditional PCR, multiplex PCR, and LAMP to distinguish insects, particularly rice planthoppers (N. lugens, S. furcifera, and L. striatellus). I think this work is interesting and useful.
Response 2: Thanks for your positive comments regarding the manuscript.
Reviewer 3 Report
The presented paper of the authors, "Development of Molecular Based Species Identification and Optimisation of Reaction Conditions for Molecular Diagnosis of Three Major Asian Planthoppers (Hemiptera: Delphacidae)", is an innovative development of a fast genetic method for identifying pests species of leafhopper of rice crops.
The main achievement of the research was to obtain three species-specific sets of primers for traditional PCR, multiplex PCR and LAMP, for species identification based on molecular material, particularly LAMP Starters, for use in the field.
Overall, this is a systematic molecular study which will add to a better understanding of the possibility to determine of three planthopper species, active rice pests. It is a useful methodological paper, and the study's stages and results are clearly presented. Moreover, the impact of the study may assist intensive field monitoring of integrated management of these species.
My opinion is based on the presented paper's general results because I do not typically specialise in molecular methods.
The gap (..........) in line 131 is accidental or not?
Author Response
Reviewer 3 comments and responses:
General Comment 1:
The presented paper of the authors, "Development of Molecular Based Species Identification and Optimisation of Reaction Conditions for Molecular Diagnosis of Three Major Asian Planthoppers (Hemiptera: Delphacidae)", is an innovative development of a fast-genetic method for identifying pest’s species of leafhopper of rice crops.
The main achievement of the research was to obtain three species-specific sets of primers for traditional PCR, multiplex PCR and LAMP, for species identification based on molecular material, particularly LAMP Starters, for use in the field.
Overall, this is a systematic molecular study which will add to a better understanding of the possibility to determine of three planthopper species, active rice pests. It is a useful methodological paper, and the study's stages and results are clearly presented. Moreover, the impact of the study may assist intensive field monitoring of integrated management of these species.
My opinion is based on the presented paper's general results because I do not typically specialize in molecular methods.
Response 1: Thanks for your positive and impressive response.
Comment 2:
The gap (..........) in line 131 is accidental or not?
Response 2: The accession numbers of three planthoppers as Laodelphax striatellus (OP895722.1), Nilaparvata lugens (OP895722.1), and Sogatella furcifera (OP895722.1- OP895724.1). We edited that in the manuscript.
Reviewer 4 Report
I think that this LAMP is useful for the rapid identification of rice planthoppers in the field or small labaratory.
2.1 sample collection
1.Please describe the collection year and month of planthopper samples.
2.Authors should cite the monograph or revisions for identifying morphological characteristics.
L198-256 please move the section 2.3.
I think that the primer design and selection is suit for the method section.
Authors should mention the varidty of LAMP and multiple PCRs in Result section.
L138 Nilaparvata is italic
L139 Laodelphax is italic
L140 Sogatella is italic
Author Response
Reviewer 4 comments and responses:
Comment 1:
2.1 sample collection: 1. Please describe the collection year and month of planthopper samples.
Response 1: Updated (June-August, 2021-22)
Comment 2
Authors should cite the monograph or revisions for identifying morphological characteristics.
L198-256 please move the section 2.3. I think that the primer design and selection is suit for the method section.
Response 2: Thank you for your kind comment. But based on our point of view, for species identification, species-specific primer design is one of the essential results. That's why we arranged that manner.
Comment 3
Authors should mention the varidty of LAMP and multiple PCRs in Result section.
Response 3: Updated as per your suggestions
Reviewer 5 Report
The manuscript "Development of Molecular Based Species Identification and 2 Optimization of Reaction Conditions for Molecular Diagnosis 3 of Three Major Asian Planthoppers (Hemiptera: Delphacidae) by M-Mafizur et al. is well written and provides information on a molecular tool to distinguish between 3 planthopper species important to the major agricultural crop, rice.
I found the manuscript easy to read and follow. However, I have a few comments/questions on the use of negative controls and limited sampling. More specifically,
1. Are there other planthopper species in the region that are not agricultural pest? I would include non-pest planthoppers in your validation of the assay. What if a common, but non-pest planthopper gives a color change for your pest species? Will costly treatments be activated based on a false positive? Along the same lines, are there other insects that could be mistaken for a planthopper that should be tested as well?
2. I worry that the sampling is too limited, and the diagnostic SNPs found will not hold up in other geographic locations. Maybe this is not important to rice in Korea, but if this tool will be used by a larger scientific community, it should hold up across other planthopper populations for other regions/countries.
Additionally, the captions description of the localities is not represented correctly in Figure S1. I only see blue dots, not red circles and triangles.
Author Response
Reviewer 5 comments and responses:
Comment 1:
I found the manuscript easy to read and follow. However, I have a few comments/questions on the use of negative controls and limited sampling. More specifically,
- Are there other planthopper species in the region that are not agricultural pest? I would include non-pest planthoppers in your validation of the assay. What if a common, but non-pest planthopper gives a color change for your pest species? Will costly treatments be activated based on a false positive? Along the same lines, are there other insects that could be mistaken for a planthopper that should be tested as well?
Response 1:
We agree with your observations and recommendations. In our investigation, the primary negative control we used was leafhoppers. We will consider your insightful comments in our future research as you stated that non-pest planthoppers with different insects should be taken into consideration as a negative control that would be more validating of our primers.
Comment 2:
- I worry that the sampling is too limited, and the diagnostic SNPs found will not hold up in other geographic locations. Maybe this is not important to rice in Korea, but if this tool will be used by a larger scientific community, it should hold up across other planthopper populations for other regions/countries.
Response 2:
We agree with your comments. In our investigation, we considered only diverse Korean local samples. As you pointed out, we need to take into account the consensus of a bigger scientific community across other planthopper populations for other countries. We will take this into consideration in our future studies.
Comment 3:
Additionally, the captions description of the localities is not represented correctly in Figure S1. I only see blue dots, not red circles and triangles.
Response 3:
Thanks for comments (unintentional errors, updated)
Round 2
Reviewer 5 Report
The authors feel my suggestions are out of the scoop of this paper, which is valid, but I would recommend that the limitations of their experimental design be discussed in the manuscript.
Author Response
Thank you for your kind comment. That's really important. Yes. No matter how much the diagnostic primers were designed based on a large number of planthoppers' sequence data available in GenBank and the diagnostic method was developed, of course, this study also has limitations. So I added that in the discussion section (lines 349-358). Thank you once again. With your valuable comments, this manuscript was polished.